# Four New *ent*-Kaurane Diterpene Glycosides from *Isodon henryi*

**DOI:** 10.3390/molecules24152736

**Published:** 2019-07-27

**Authors:** Ya-Lin Liu, Ling-Xia Zhang, Hong Wu, Sui-Qing Chen, Jun Li, Li-Ping Dai, Zhi-Min Wang

**Affiliations:** 1School of Pharmacy, Henan University of Traditional Chinese Medicine, Zhengzhou 450046, China; 2Research Center for Classic Chinese Medines & Health Herbal Products, Zhengzhou 450046, China; 3Laboratory of Cell Imaging, Henan University of Chinese Medicine, Zhengzhou 450002, China; 4Institute of Chinese Materia Medica, China Academy of Chinese Medical Sciences, Beijing 100700, China

**Keywords:** *Isodon henryi*, *ent*-kaurane diterpene, cytotoxic activity

## Abstract

To obtain diterpene glycosides from an aqueous extract of the aerial parts of *Isodon henryi* and further investigate their cytotoxicities, in this study, a total of seven compounds were isolated, including six *ent*-kaurane diterpene glycosides (**1**–**6**) and one diterpene aglycon (**7**). Among the seven *ent*-kaurane diterpenes obtained, four were novel compounds, including *ent*-7,20-epoxy- kaur-16-en-1*α*,6*β*,7*β*,15*β*-tetrahydroxyl-11-*O*-*β*-d-glucopyranoside (**1**), *ent*-7,20-epoxy-kaur-16-en- 6*β*,7*β*,14*β*,15*β*-tetrahydroxyl-1-*O*-*β*-d-glucopyranoside (**2**), *ent*-7,20-epoxy-kaur-16-en-6*β*,7*β*,15*β*- trihydroxyl-1-*O*-*β*-d-glucopyranoside (**3**), and *ent*-7,20-epoxy-kaur-16-en-7*β*,11*β*,14*α*,15*β*-tetrahydr- oxyl-6-*O*-*β*-d-glucopyranoside (**4**), and three were isolated from this plant for the first time (**5**–**7**). Their structures were elucidated by utilizing spectroscopic methods and electronic circular dichroism analyses. Furthermore, the cytotoxicities of all seven compounds were investigated in four human cancer cell lines, including A2780, BGC-823, HCT-116, and HepG2. The IC_50_ values of these diterpenes ranged from 0.18 to 2.44 mM in the tested cell lines. In addition, the structure–cytotoxicity relationship of diterpene glycosides was also evaluated to study the effect of glycosylation on the cytotoxicity of diterpene compounds.

## 1. Introduction

Diterpenoids from *Isodon* plants possess potent cytotoxic activities and are potential candidates for the treatment of cancer [1,2,3,4,5,6,7,8]. Since the 1980s, an increasing number of investigations have focused on the cytotoxicities of diterpenoids from *Isodon* plants [9,10,11,12,13,14]. In the past five years, we have performed continuous studies on the cytotoxicities of phytochemicals, including structure–activity relationship analyses of *Isodon* plants, in order to discover antitumor dominant compounds. Our previous studies indicated that a cyclopentanone conjugated with an exomethylene group is one of the leading activity-related factors, and also validated the cytotoxicities of the 7,20-non-epoxy group of *ent*-kaurane diterpenoids with the 20-OAc group [15]. Accordingly, we wondered about the structure–cytotoxicity relationship of diterpene glycosides. As is known, only a few diterpenoid glycosides have been reported from *Isodon* plants. Additionally, there are few studies that have analyzed the structure–activity relationship of diterpenoid glycosides and the corresponding aglycones. The directional separation of *ent*-kaurane glycosides is a key issue that can be drawn upon to explain the effects of glycosidic bonds on cytotoxicity.

To investigate glycoside-caused effects on cytotoxicity, the separation and preparation of diterpene glycosides were firstly performed by TLC tracing. A total of seven diterpenoids, including six diterpenoid glycosides, were isolated from an aqueous extract of the aerial parts of *I. henryi* for the first time. Among the seven compounds, four were novel compounds (**1**–**4**) and three were isolated from this plant for the first time (**5**–**7**). Subsequently, the cytotoxicity of each compound was evaluated. Interestingly, Compound **7** is just the aglycone of Compounds **2** and **6**. Accordingly, structure–activity relationship analyses of these two diterpenoid glycosides and their common aglycone were also conducted. Structures of Compounds **1**–**7** as shown in Figure 1.

## 2. Results and Discussion

### 2.1. Identification of New Compounds

Compound **1** was isolated as an amorphous powder. The molecular formula of **1** was determined to be C_26_H_40_O_11_ by HR-ESI-MS at *m*/*z* 551.2479 [M + Na]^+^ (calculated for C_26_H_40_O_11_Na^+^, 551.2472), indicating seven degrees of unsaturation. The UV spectrum of **1** showed an absorption maximum at 195 nm. Its IR spectrum displayed the absorption bands of hydroxyl (3425 cm^−1^) and ether (1065 cm^−1^) groups. The ^1^H NMR data of **1** (Table 1), along with its HSQC spectrum, displayed characteristic resonances for two methyl groups (*δ*_H_ 0.99 (s) and 1.09 (s)), an oxygenated methylene group (*δ*_H_ 4.01 (dd, 10.2, 2.2) and 4.16 (d, 10.2)), an olefinic methylene group (*δ*_H_ 4.96 (br s) and 5.04 (d, 2.1)), and four oxygenated methine protons (*δ*_H_ 4.12 (dd, 11.7, 5.6), 3.65 (d, 4.9), 4.52 (overlap), and 4.54 (overlap)). In addition, the characteristic signal of an anomeric proton at *δ*_H_ 4.36 (d, 7.8), along with six other protons ranging from *δ*_H_ 3.19 to 3.83, suggested the presence of a sugar residue. The ^13^C NMR and DEPT spectra of **1** (Table 1) exhibited the presence of 26 carbon resonances corresponding to two methyls, seven methylenes (one olefinic and two oxygenated), 12 methines (nine oxygenated), four quaternary carbons (one olefinic), and an oxygenated secondary carbon. Among these carbons, six were associated with the sugar moiety. A comparison of the aforementioned spectroscopic data of **1** with those of isodonterpene II [16] revealed that Compound **1** may be a diterpenoid glycoside possessing the same *ent*-kaurane carbon skeleton as is seen in **5**.

The above deduction was fully supported by the ^1^H–^1^H COSY correlations of H-1/H_2_-2/H_2_-3, H-5/H-6, and H-9/H-11/H_2_-12/H-13/H_2_-14, and key HMBC correlations from H_3_-18 to C-3/C-4/C-5/C-19, from H_2_-20 to C-1/C-5/C-6/C-9/C-7, from H_2_-17 to C-13/C-16/C-15, and from H_2_-14 to C-7/C-8/C-9/C-15 (Figure 2). Moreover, the key HMBC correlations from the two overlapped protons H-15/H-11 to C-1′ and from H-1′ to the two overlapped carbons C-15/C-11, combined with the NOESY correlations of H-1′/H-9, H-1′/H_2_-12, and H-1′/-11 (Figure 3), indicated that the sugar residue was located at C-11. In order to further confirm the structure of the sugar residue, acidic hydrolysis of **1** was carried out. d-Glucose was identified by derivatization with l-cysteine methyl ester hydrochloride and *o*-Tolyl isothiocyanate, followed by HPLC analysis [17]. Therefore, the planar structure of **1** was determined, as shown in Figure 1.

The relative configuration of **1** was established by the NOESY spectrum (Figure 3) and the coupling constants. The key NOESY correlations of H-1/H-5, H-1/H-9, H-11/H_2_-20, Me-19/H_1_-20, Me-18/H-5, H_1_-12/H-9, H-11/H-14, and H-15/H-14 were observed, but no correlations of H-9/H-11 and H-9/H-15 were found. This information confirmed the relative configuration of the diterpenoid unit, as shown in Figure 3. The large coupling constant (*J* = 7.8 Hz) of the anomeric proton demonstrated that the glucose exhibited a *β*-orientation. Finally, the absolute configuration of the diterpenoid unit was determined as *1S*, *5R*, *6S*, *7S*, *8S*, *9S*, *10S*, *11S*, *13S*, and *15R* based on TDDFT calculations (Figure 4). Therefore, the structure of **1** was determined to be *ent*-7,20-epoxy-kaur- 16-en-1*α*,6*β*,7*β*,15*β*- tetrahydroxyl-11-*O*-*β*-d-glucopyranoside.

Compound **2** was isolated as an amorphous powder. It has the same molecular formula as **1**, which was determined to be C_26_H_40_O_11_ on the basis of HR-ESI-MS at *m*/*z* 551.2479 [M+Na]^+^ (calculated for C_26_H_40_O_11_Na^+^, 551.2472). The similarities of its UV, IR, and NMR (Table 1) data with those of **1** indicated that these compounds were structural analogues. A detailed comparison of the NMR data of **2** with those of **1** revealed that the C-14 signal of **2** was shifted downfield from *δ*_C_ 30.1 to *δ*_C_ 75.8, and the C-11 signal of **2** was shifted upfield from *δ*_C_ 75.1 to *δ*_C_ 19.4, which indicated that a methylene group at C-14 and a sec-alcohol group at C-11 in **1** replace each other in **2**. This deduction was further corroborated by the ^1^H–^1^H COSY (Figure 2) correlations of H-9/H_2_-11/H_2_-12/H-13/H-14. Moreover, the key HMBC (Figure 2) correlations from H-1 to C-1′ and from H-1′ to C-1 indicated that the sugar residue was located at C-1. It was further certified to be d-glucose through an acidic hydrolysis experiment. The relative configuration of **2** was deduced from its NOESY spectrum. NOESY crosspeaks (Figure 3) were observed between H-5/H-9, H_3_-19/H_2_-20, and H-1/H_1_-20, and H_1_-20/H-14, H_2_-12/H-14, H-14/H_1_-20, and H-6/H_3_-19, but no correlations between H-1/H-5, H-14/H-15, and H-9/H-15 were found. H-1, H_3_-19, H_2_-20, H-14, H-13, and H-6 were therefore assigned as the same orientation whereas H-5, H-9, and H-15 had another orientation. In addition, Compound **2** exhibited almost the same ECD absorption (Figure 4) as that of **1**. Hence, Compound **2** was denominated to be *ent*-7,20-epoxy-kaur-16-en-6*β*,7*β*,14*β*,15*β*-tetrahydroxyl-1-*O*-*β*-d- glucopyranoside.

Compound **3** was also isolated as an amorphous powder. The molecular formula of **3** was determined to be C_26_H_40_O_10_ on the basis of positive HR-ESI-MS at *m*/*z* 535.2505 [M+Na]^+^ (calculated for C_26_H_40_O_10_Na^+^, 535.2513). The UV spectrum of **3** showed an absorption maximum at 195 nm. Its IR spectrum displayed the absorption bands of a hydroxyl group (3409 cm^−1^) and ether group (1075 cm^−1^). The ^1^H and ^13^C NMR spectra showed close similarities to those of **1** (Table 1), except for an sp3 methylene at C-11 (*δ*_C_ 19.7) in **3** being replaced by an oxymethine (*δ*_C_ 75.1) in **1**. These results were confirmed by HMBC and ^1^H–^1^H COSY experiments (Figure 2). The key HMBC correlations from H-1 to C-1′ and from H-1′ to C-1 and the same acid hydrolysis experiment as that of **1** indicated that a d-glucose residue was located at C-1. Finally, similar NOESY (Figure 3) and ECD (Figure 4) data showed that **3** possessed the same relative configuration and absolute configuration as **1**. Therefore, the structure of **3** was defined as *ent*-7,20-epoxy-kaur-16-en-6*β*,7*β*,15*β*-trihydroxyl- 1-*O*-*β*-d-glucopyranoside.

Compound **4** was obtained as a white, amorphous powder, and its molecular formula C_26_H_40_O_11_ (seven indices of hydrogen deficiency) was deduced based upon the HR-ESI-MS analysis (*m*/*z* 551.2527 [M+Na]^+^ calculated for C_26_H_40_O_11_Na^+^, 551.2521). The UV spectrum of **4** showed an absorption maximum at 195 nm and its IR spectrum displayed the absorption bands of a hydroxyl group (3410 cm^−1^) and ether group (1076 cm^−1^). Analysis of the 1D and 2D NMR spectra indicated that the structure of **4** was similar to that of **2**, except for a methylene group at C-1 (*δ*_C_ 32.2) and a sec-alcohol group at C-11 (*δ*_C_ 61.7) in **4** (Table 1). The HMBC (Figure 2) correlations from H_2_-20 to C-1 and ^1^H–^1^H COSY (Figure 2) correlations of H-9/H-11/H_2_-12/H-13/H-14 and H_2_-1/H_2_-2/H_2_-3 further supported the deduction. Moreover, a d-glucose residue was located at C-6, as supported by the key HMBC correlation from H-6 to C-1′ and from H-1′ to C-6 and the acid hydrolysis experiment. Additionally, similar NOESY (Figure 3) data showed that **4** possessed the same relative configuration as **2**, except that the hydroxyl group at C-11 was determined to be *β*-oriented based on NOESY correlations of H-11/H-14. In addition, Compound **4** exhibited almost the same ECD absorption (Figure 4) as that of **2**. Therefore, Compound **4** was denominated to be *ent*-7,20-epoxy-kaur-16-en-7*β*,11*β*,14*α*,15*β*-tetrahydroxyl-6-*O*-*β*-d-glucopyranoside.

Through NMR data analysis and a comparison with the reported spectroscopic data, three known compounds (**5**–**7**) were identified as isodonterpene II (**5**) [16], enmenol-1-*β*-glucoside (**6**) [18], and enmenol (**7**) [19].

### 2.2. Cytotoxicity Assay

Using the MTT method [20,21], all compounds were evaluated for their cytotoxic effects against four human cancer cell lines, including A2780, BGC-823, HCT-116, and HepG2. Most of the tested compounds exhibited potent cytotoxicities. The results are presented in Table 2.

### 2.3. Analysis of Structure–Activity Relationships

The structure–activity relationships of seven compounds obtained from *I. henryi* were assessed based on the results of a cytotoxic activity test. Compounds **1**–**6** were 7,20-epoxy *ent*-kaurane glycosides. Additionally, the ketone carbonyl groups at the 15 position were all reduced to hydroxyl groups (*α*,*β*-unsaturated pentanone, disappearing). The difference between Compounds **1** and **5** is the location of the linking sugar, and the structural difference between Compounds **2** and **6** is the orientation of glucose, where the glucose had a *β*-orientation in Compound **2**. Compounds **5** and **6** have very similar structures, except that **6** has an additional hydroxyl group at the 11 position. It is worth noting is that Compound **7** is the aglycone of 6. However, the cytotoxicity results showed that there was no significant difference in cytotoxicity between the seven compounds. By further comparing the cytotoxic activity of the seven compounds and oridonin (7,20-epoxy kaurane diterpenoid composed of α,β-unsaturated pentone and exocyclic methylene), it was found that the cytotoxic activity was significantly weaker than oridonin [15]. The above results indicate that the introduction of a glycosyl group has no significant effect on cytotoxic activity. Additionally, the cytotoxic activity was significantly reduced without *α*,*β*-unsaturated pentones and exocyclic methylene groups in the structure of 7,20 epoxy *ent*-kaurane diterpenoids.

## 3. Experimental Section

### 3.1. General Information

IR spectra were recorded on a Spectrum Nicolet iS5 Spectrometer (Thermo Fisher Scientific, Madison, WI, USA). UV spectra were recorded on a Shimadzu double-beam 210A spectrophotometer (Shimadzu, Kyoto, Japan). Optical rotation was measured using a SEPA-300 polarimeter (Horiba, Tokyo, Japan). NMR spectra were recorded on a Bruker Avance III spectrometer (Bruker, Billerica, Germany) with TMS as the internal standard. Chemical shifts (*δ*) are expressed in ppm with reference to the solvent signals. HR-ESI-MS data was acquired using an Acquity UPLC I-Class (Waters, Acquity UPLC I-Class/Xevo G2-XS, QT, USA). Analytical HPLC was performed on a C_18_ column (Gemini, 4.6 × 150 mm, 5 μm) with UV detection at 250 nm. Semi-preparative HPLC was performed on a Waters 600/Waters 2487 (Waters, Milford, MA, USA) with a YMC (250 mm × 10 mm I.D. 5 μm) column. Column chromatography was performed on silica gel (100–200 mesh and 200–300 mesh, Qingdao Marine Chemical Inc., Qingdao, China). Solvents were distilled prior to use. Spectroscopic grade solvents were used. TLC was carried out on precoated silica gel GF254 plates. Spots were visualized by heating silica gel plates sprayed with 10% H_2_SO_4_ in ethanol (*v*/*v*).

### 3.2. Plant Material

The aerial parts of *I. henryi* were collected from Luanchuan County in Henan Province, China, in September 2016 and authenticated by Professor Xiao-Zheng Luo of the Henan University of Chinese Medicine. A voucher specimen (No. 2016-0906) was deposited in the Key Laboratory of Traditional Chinese Medicine Chemistry and Resource of Henan Province.

### 3.3. Extraction and Isolation

The air-dried and powdered aerial parts of *I. henryi* (20 kg) were decocted three times with water (320 L×1.5 h) at 100 °C. The decoction was then evaporated to obtain a concentrate (1.5 kg). Then, ethyl acetate and *n*-butanol were used for extraction. The solvent was recovered by vacuum distillation, to obtain two fractions (Fr. A–B).

Fr. B (75.0 g, *n*-butanol extract) was subjected to column chromatography over silica gel (100–200 mesh), eluted with DCM–MeOH (40:1, 30:1, 20:1, 10:1, 5:1, 0:1) to yield six sub-fractions (Fr. B1–6). Fr. B5 (8.8 g) was again subjected to column chromatography over silica gel (100–200 mesh) and eluted with DCM–MeOH (50:1, 20:1, 10:1, 5:1, 0:1) to yield three sub-fractions Fr. B5a–c. Fr. B5b (1.5 g) was further purified by semi-preparative HPLC (ACN–H_2_O, 18:82, 2.5 mL/min). Detection was monitored at 190 and 230 nm. Compounds **1** (6 mg), **5** (10 mg), **2** (5 mg), **3** (6 mg), and **4** (5 mg) were obtained at 23.5, 24.3, 26.6, 30.2, and 39.7 min, respectively. Compounds **6** (20 mg) and **7** (15 mg) were purified after repeated chromatography over silica gel from the DCM–MeOH (5:1) fraction of Fr. B5c and the DCM–MeOH (20:1) fraction of Fr. B6.

The following is a list of all the results obtained of *I. henryi*.

*ent*-7,20-epoxy-kaur-16-en-1*α*,6*β*,7*β*,15*β*-tetrahydroxyl-11-*O*-*β*-d-glucopyranoside (**1**): amorphous powder (MeOH); ${\left[\alpha \right]}_{D}^{20}$ = −10.49 (c 0.11, MeOH), IR (KBr) λmax (cm^−1^): 3425, 2921, 1345, 1149, 1129, 1103, 1065, 1018 cm^−1^; HR-ESI-MS *m*/*z*: 551.2479 [M+Na]^+^ (calculated for C_26_H_40_O_11_Na^+^, 551.2472); ECD (MeOH) λmax (Δε) 206 (+3.84) nm; See Table 1 and Appendix A for ^1^H NMR (MeOH, 500 MHz) and ^13^C NMR (MeOH, 125 MHz) spectral data.

The acid hydrolysis and sugar analysis of the isolated Compound **1** were performed as previously described in the literature [17]. Compound **1** (1 mg) was heated with L-cysteine methyl ester (1 mL) in pyridine at 60 °C for 60 min, and *o*-Tolyl isothiocyanate (1 mL) was then added to the reaction mixture and further reacted at 60 °C for 60 min. The reaction mixture (10 μL) was analyzed by analytical HPLC and eluted with 20% aqueous CH_3_CN containing 0.1% TFA at a flow rate of 1 mL/min over a 40 min run. d-Glucose (*t_R_* = 17.13 min) was identified by comparing retention times with those of the authentic samples (*t_R_* = 17.12 min).

*ent*-7,20-epoxy-kaur-16-en-6*β*,7*β*,14*β*,15*β*-tetrahydroxyl-1-*O*-*β*-d-glucopyranoside (**2**): amorphous powder (MeOH); ${\left[\alpha \right]}_{D}^{20}$ = −13.94 (c 0.35, MeOH), IR (KBr) λmax (cm^−1^): 3402, 2944, 2871, 2521, 1391, 1370, 1212, 1070, 1028, 974 cm^−1^; HR-ESI–MS *m*/*z*: 551.2479 [M+Na]^+^ (calculated for C_26_H_40_O_11_Na^+^, 551.2472); ECD (MeOH) λmax (Δε) 205 (+16.32) nm; See Table 1 for ^1^H NMR (MeOH, 500 MHz) and ^13^C NMR (MeOH, 125 MHz) spectral data.

*ent*-7,20-epoxy-kaur-16-en-6*β*,7*β*,15*β*-trihydroxyl-1-*O*-*β*-d-glucopyranoside (**3**): amorphous powder (MeOH); ${\left[\alpha \right]}_{D}^{20}$ = −19.36 (c 0.06, MeOH), IR (KBr) λmax (cm^−1^): 3409, 2929, 1454, 1372, 1353, 1075, 1036, 947 cm^−1^; HR-ESI-MS *m*/*z*: 535.2505 [M+Na]+ (calculated for C_26_H_40_O_10_Na^+^, 535.2513); ECD (MeOH) λmax (Δε) 205 (+7.16) nm; See Table 2 for ^1^H NMR (MeOH, 500 MHz) and ^13^C NMR (MeOH, 125 MHz) spectral data.

*ent*-7,20-epoxy-kaur-16-en-7*β*,11*β*,14*α*,15*β*-tetrahydroxyl-6-*O*-*β*-d-glucopyranoside (**4**): amorphous powder (MeOH); ${\left[\alpha \right]}_{D}^{20}$ = −32.93 (c 0.30, MeOH), IR (KBr) λmax (cm^−1^): 3410, 2927, 1461, 1352, 1075, 1035, 971 cm^−1^; HR-ESI-MS *m*/*z*: 551.2527 [M+Na]^+^ (calculated for C_26_H_40_O_11_Na^+^, 551.2521); ECD (MeOH) λmax (Δε) 205 (+17.78) nm; See Table 2 for ^1^H NMR (MeOH, 500 MHz) and ^13^C NMR (MeOH, 125 MHz) spectral data.

### 3.4. Cytotoxicity Assay

Four human cancer cell lines (ovarian cancer cell line A2780, gastric cancer cell line BGC-823, colon carcinoma cell line HCT-116, and hepatic cancer cell line HepG2) were used. All cells were cultured in RPMI-1640 medium supplemented with 10% fetal bovine serum in a humidified atmosphere with 5% CO_2_ at 37 °C. The cytotoxicity assay was performed according to the MTT (3-(4,5-dimethylthiazol-2-yl) -2,5-diphenyl tetrazolium bromide) method using 96-well microplates [22]. Briefly, the cells were cultured in RPMI-1640 medium supplemented with 10% fetal bovine serum in a humidified atmosphere with 5% CO_2_ at 37 °C. Next, 100 μL of adherent cells at a density of 5 × 10^4^ cells/mL were seeded into each well of the 96-well cell culture plates and incubated in 5% CO_2_ at 37 °C for 24 h to form a monolayer on the flat bottom. Next, the supernatant of each well was removed, after which 100 μL of fresh medium and 100 μL of medium containing a test sample were added to the well. The plate was then incubated in 5% CO_2_ at 37 °C for 24 h. Next, 20 μL of 5 mg/mL MTT in DMSO was added to each well and further incubated for 4 h. The supernatant was carefully removed from each well and 150 μL of DMSO was added. The plate was then vortex-shaken for 15 min to dissolve the blue formazan crystals. The optical density (OD) of each well was measured on a Genois microplate reader (Tecan GENios, Männedorf, Switzerland) at a wavelength of 570 nm.

For the evaluation of cytotoxicity, each tumor cell line was exposed to the test compound at concentrations of 2 × 10^−5^, 2 × 10^−6^, and 2 × 10^−7^ mol/L. The inhibitory rate of cell growth was calculated according to the following formula: Inhibition rate (%) = (OD_control_ − OD_treated_)/OD_control_ × 100. Finally, IC_50_ values were calculated using SPSS 16.0 statistical software.

## 4. Conclusions

Phytochemical investigations on a water extract of the aerial parts of *I. henryi* resulted in the isolation of four new compounds **1**–**4**, along with three known compounds **5**–**7**. The isolated compounds were identified as *ent*-7,20-epoxy-kaur-16-en-1*α*,6*β*,7*β*,15*β*-tetrahydroxyl-11-*O*-*β*-d- glucopyranoside (**1**), *ent*-7,20-epoxy-kaur-16-en-6*β*,7*β*,14*β*,15*β*-tetrahydroxyl-1-*O*-*β*-d-glucopyrano- side (**2**), *ent*-7,20-epoxy-kaur-16-en-6*β*,7*β*,15*β*-trihydroxyl-1-*O*-*β*-d-glucopyranoside (**3**), and *ent*-7,20- epoxy-kaur-16-en-7*β*,11*β*,14*α*,15*β*-tetrahydroxyl-6-*O*-*β*-d-glucopyranoside (**4**). Compounds **5**, **6**, and **7** were isolated from *I. henryi* for the first time. All of the compounds were evaluated for their cytotoxic effects against four human tumor cell lines (A2780, BGC-823, HCT-116 and HepG2) and all showed certain cytotoxic activity against the four types of human tumor cells tested.

Cytotoxicity results showed that the introduction of diterpene glycosides led to no significant cytotoxic activity. Compounds **1**–**7** displayed cytotoxicity activity in the tested cell lines, with IC_50_ values ranging from 0.18 to 2.44 mM, and it was found that the difference in the position of the diterpene glycosides had no effect on the cytotoxic activity. Therefore, it is worth studying the directed separation and cytotoxic activity of 7,20 epoxy *ent*-kaurane glycosides with *α,β*-unsaturated pentanone.

## Figures and Tables

**Figure 1 molecules-24-02736-f001:**
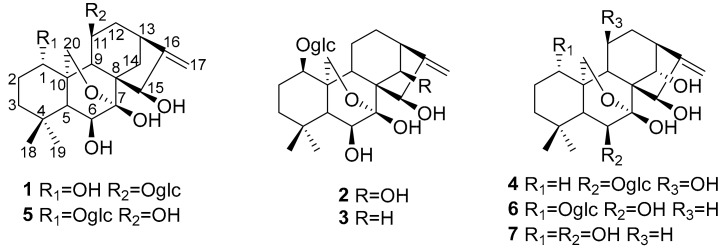
Structures of Compounds **1**–**7**.

**Figure 2 molecules-24-02736-f002:**
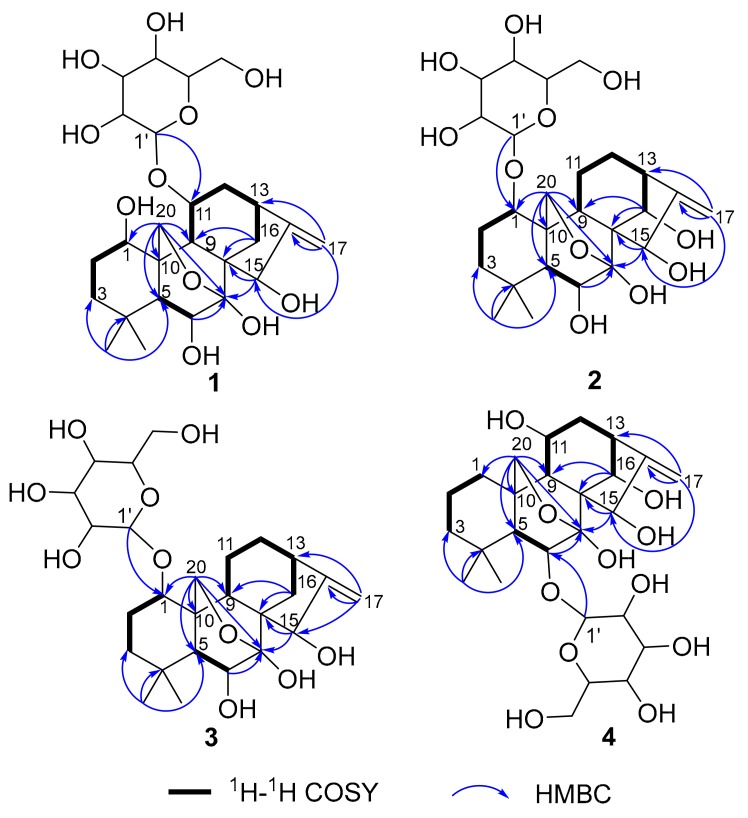
Key HMBC and ^1^H–^1^H COSY correlations for Compounds **1**–**4**.

**Figure 3 molecules-24-02736-f003:**
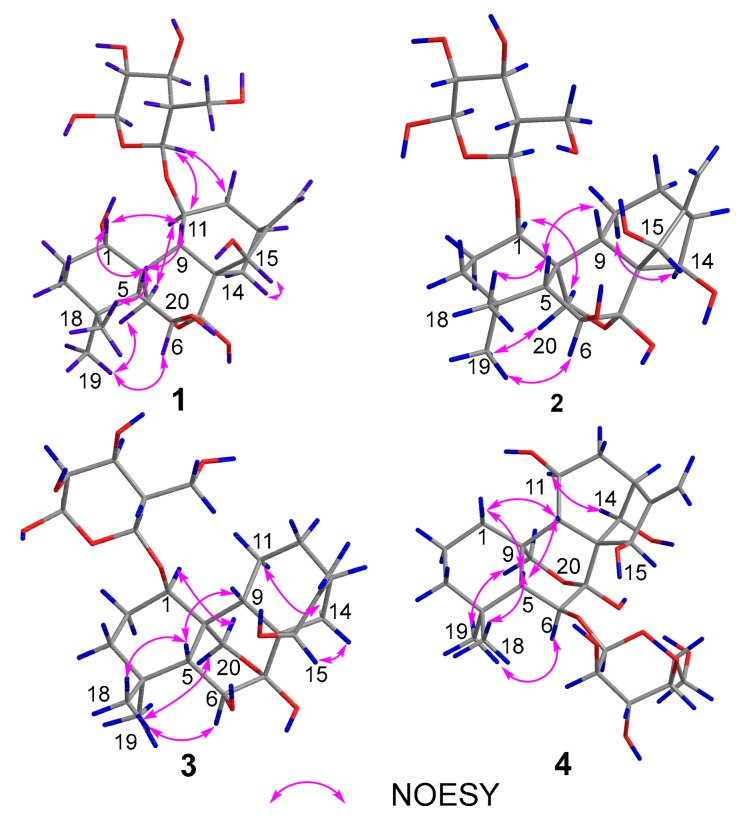
Key NOESY correlations of Compound **1**–**4**.

**Figure 4 molecules-24-02736-f004:**
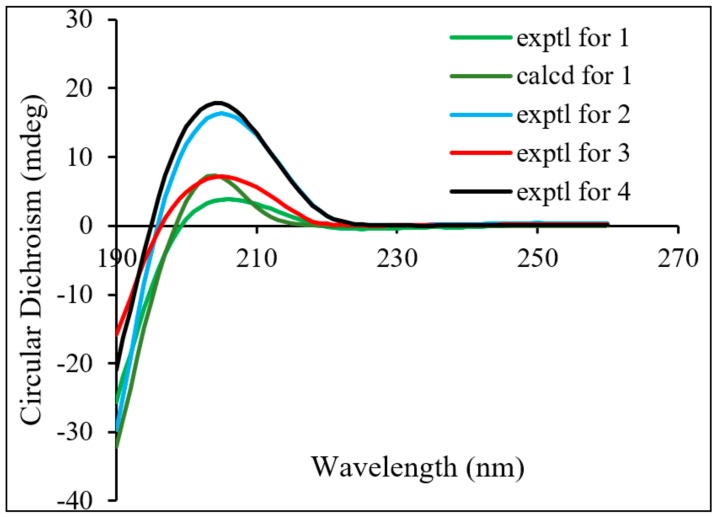
Calculated ECD spectra of **1** and experimental ECD spectra of **1**–**4**.

**Table 1 molecules-24-02736-t001:** ^1^H and ^13^C NMR data of Compounds **1**–**4** in CD_3_OD (500 and 125 *MHz δ* in ppm).

No.	1		2		3		4	
	*δ* _C_	*δ*_H_ (*J* in Hz)	*δ* _C_	*δ*_H_ (*J* in Hz)	*δ* _C_	*δ*_H_ (*J* in Hz)	*δ* _C_	*δ*_H_ (*J* in Hz)
1	72.1	4.12, dd (11.7, 5.6)	72.3	3.85, overlap	85.4	3.57, dd (11.4, 5.6)	32.2	1.58, m
2	28.5	1.59, overlap	31.4	a 1.26, m b 1.83, overlap	28.9	1.66, m 2.23, overlap	19.6	1.49, overlap
3	40.1	a 1.33, m b 1.45, m	42.2	a 1.22, overlap b 1.43, overlap	39.8	a 1.27, overlap b 1.43, dt (13.5, 3.6)	43.2	a 1.19, m b 1.43, overlap
4	34.7		34.8		34.2		35.2	
5	60.1	1.19, d (4.9)	59.2	1.21, d (4.9)	59.7	1.29, overlap	57.8	1.33, d (5.3)
6	74.7	3.65, d (4.9)	73.6	3.63, d (4.9)	75.1	3.69, d (6.0)	76.2	4.22, d (5.3)
7	97.4		100.1		97.4		101.2	
8	53.5		54.4		53.1		54.7	
9	49.2	2.33, d (8.7)	50.3	2.47, overlap	44.5	2.04, dd (13.0, 4.7)	51.2	2.28, dd (9.4, 2.1)
10	42.4		38.4		42.2		38.3	
11	75.1	4.52, overlap	19.4	1.48, overlap	19.7	a 1.55, overlap b 2.04, qd (13.0, 8.2)	61.7	3.88, overlap
12	41.1	a 1.82, dd (14.4, 8.7) b 2.76, dt (14.4, 9.3)	44.0	a 1.83, overlap b 2.98, dt (14.8, 9.0)	33.2	a 1.34, td (12.6, 7.1) b 2.21, overlap	43.7	a 1.51, overlap b 2.72, dt (13.9, 9.2)
13	38.2	2.61, dd (9.8, 4.3)	47.4	2.48, overlap	38.0	2.56, dd (9.5, 4.5)	46.8	2.49, dd (9.3)
14	30.1	1.64, overlap	75.8	4.29, s	26.7	a 1.52, overlap b 1.75, d (12.0)	77.0	4.38, s
15	75.1	4.54, overlap	73.3	4.96, t (2.6)	75.6	4.43, t (2.6)	72.6	5.02, t (2.3)
16	160.8		158.8		162.1		160.4	
17	107.2	a 4.96, br s b 5.04, d (2.1)	110.3	a 5.19, d (2.6) b 5.21, br s	107.3	a 4.97, t (2.6) b 5.02, br d (2.6)	110.5	a 5.17, br d (2.3) b 5.29, br s
18	32.8	0.99, s	33.5	1.01, s	33.1	1.01, s	33.6	1.07, s
19	22.1	1.09, s	22.6	1.09, s	22.1	1.12, s	23.1	1.12, s
20	64.9	a 4.01, dd (10.2, 2.2) b 4.16, d (10.2);	67.7	a 3.83, overlap b 4.08, dd (9.9, 2.4)	64.3	a 3.97, dd (11.9, 1.4) b 4.32, overlap	67.3	a 3.85, d (9.9) b 4.10, dd (9.9, 2.1)
1′	104.1	4.36, d (7.8)	104.5	4.25, d (7.7)	104.8	4.32, d (7.7)	102.0	5.09, d (7.9)
2′	75.7	3.19, t (7.8)	75.5	3.12, dd (9.1, 7.7)	75.6	3.13, dd (9.0, 7.7)	76.0	3.11, dd (9.2, 7.9)
3′	78.9	3.28, overlap	78.1	3.29, overlap	78.5	3.31, overlap	78.5	3.35, overlap
4′	71.4	3.26, t (10.1)	71.8	3.23, overlap	71.6	3.24, overlap	77.0	3.24, overlap
5′	77.8	3.23, overlap	77.8	3.23, overlap	77.7	3.23, overlap	78.0	3.28, overlap
6′	62.8	a 3.65, m b 3.83, dd (11.9, 2.0)	63.0	a 3.64, overlap b 3.87, overlap	62.8	a 3.62, dd (11.9, 5.4) b 3.82, dd (11.9, 2.1)	63.3	a 3.68, dd (11.5, 5.7) b 3.91, overlap

**Table 2 molecules-24-02736-t002:** Cytotoxic activities (IC_50_, mM) of all tested compounds on four human cancer cell lines.

Sample	A2780	BGC-823	HCT-116	HepG2
**1**	0.53 ± 0.02	1.15 ± 0.04	0.38 ± 0.01	0.96 ± 0.06
**2**	0.53 ± 0.04	0.99 ± 0.05	0.35 ± 0.01	0.89 ± 0.03
**3**	0.27 ± 0.02	0.87 ± 0.03	0.26 ± 0.01	0.21 ± 0.04
**4**	0.60 ± 0.01	2.44 ± 0.08	0.29 ± 0.02	0.61 ± 0.11
**5**	0.24 ± 0.03	0.85 ± 0.07	0.28 ± 0.03	0.23 ± 0.05
**6**	0.68 ± 0.01	0.86 ± 0.02	0.40 ± 0.01	0.65 ± 0.05
**7**	0.28 ± 0.04	0.38 ± 2.11	0.29 ± 0.45	0.18 ± 4.42
DDP	0.002 ± 0.02	0.02 ± 0.14	0.01 ± 0.05	0.01 ± 0.01

DDP (cisplatin) was used aspositive controls.

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
