# Peer review of "Four New *ent*-Kaurane Diterpene Glycosides from *Isodon henryi"

_molecules, 2019, doi:10.3390/molecules24152736_

Round 1
Reviewer 1 Report
This paper describes the structural elucidation of four new glycoside diterpenes from Isodon henryi (Lamiaceae). Some mistakes were make in the paper and should be corrected. Structures should be carrefully checked, especially the compound 2.
At line 67: only six carbons are related to sugar moiety.
At lines 73-74: the authors make a mistake. The sugar moiety was bonded at C-11 (Figure 1) however in the text at C-15. The signals of H-11 and H15 are overlapped. What is the correct position of sugar moiety ?
At line 94: "C-11 signal of 2 shifted toward upfield from δC 19.4 to δC 75.1". The information should be corrected to C-11 signal of 2 shifted toward upfield from δC 75.1 to δC 19.4.
At line 105: the authors named the compound 2 as ...... 6β,7β,14α,15β-tetrahydroxyl ..... . However, in structure on Figure 1 the hydroxyl group at C-14 was represented with beta-configuration.
Compound 2 has chemical shift of C-1 at δ 72.3. However, compound 3 has the same carbon atom with chemical shift at δ 85.4 . Justify this larger difference.
Why the glucose moiety is not bonded to C-14 in structure 2?
Please, verify carefully the HMBC correlations spectra of compound 2. Analyzing the supplementary material I think that structure 2 is wrong.
Check carefully the given names for compounds.
Despite the reference cited in acid hydrolysis, the methodology employed should be summarized and added in experimental section.
Author Response
Dear Professor:
Thank you very much for your letter and giving us opportunity to revise the manuscript entitled “Four new ent-Kaurane diterpene glycosides from Isodon henryi” (molecules-519355). We have carefully studied the reviewer’s critical comments and thoughtful suggestions. Based on the comments we received, careful modifications have been made to the manuscript. All changes made to the text are in red so that they may be easily identified. If there are more questions, we are willing to revise it again.
Below you will find our point-by-point responses to the reviewers’ comments.
Thank you and all the reviewers again for the kind and helpful advice.
A list of changes and responses to reviewers are as follows:
Reviewer 1:
Q1: Structures should be carefully checked, especially the compound 2. Please, verify carefully the HMBC correlations spectra of compound 2. Analyzing the supplementary material I think that structure 2 is. Compound 2 has chemical shift of C-1 at δC 72.3. However, compound 3 wrong has the same carbon atom with chemical shift at δC 85.4. Justify this larger difference.
A1: Thanks. We have checked the structure of compound 2 following your suggestion. Based on further comprehensive analysis of the nuclear magnetic data, we think that it could be still the previous structure. Some key data were presented as follow shown in Figure 1-5. However, we found the difference of chemical shift at C-1 between compound 2 and 3. According to the difference of chemical shift, the difference may be due to the configuration of carbon at C-1. But the NOESY correlations (Figure 6) of H-1/H-20a and no correlations of H-1/H-5 or H-1/H-9, indicated that the H-1 was α-orientation, possessing the same configuration with compound of 3. Structurally, the mainly difference of two compounds is substituent group at C-14. Hence, we think it is possible that the hydroxyl group in compound 3 caused the electron cloud density of the carbon at C-1 to decrease and thus the chemical shift toward downfield.
Figure 1 Selected key 1H-1H COSY data of compound 2
Figure 2 Selected key 1H-1H COSY data of compound 2
Figure 3 Selected key HMBC data of compound 2
Figure 4 Selected key HMBC data of compound 2
Figure 5 Selected key HMBC data of compound 2
Figure 6 Selected key NOESY data of compound 2
Q2: At line 67: only six carbons are related to sugar moiety.
A2: We have corrected the mistake. Thank you.
Q3: At lines 73-74: the authors make a mistake. The sugar moiety was bonded at C-11 (Figure 1) however in the text at C-15. The signals of H-11 and H-15 are overlapped. What is the correct position of sugar moiety ?
A3: Firstly, the key HMBC correlations from the two overlapped protons H-15/H-11 to C-1’ and from H-1’ to the two overlapped carbons C-15/C-11 indicated that the sugar residue was located at C-11 or C-15. Secondly, the NOESY correlations of H-1’/H-9, H-1’/H-12b, and H-1’/-11 indicated that it was located at C-11, rather than C-15. Thank you for your advice. We have revised it in the revision.
Q4: At line 94: "C-11 signal of 2 shifted toward upfield from δC 19.4 to δC 75.1". The information should be corrected to C-11 signal of 2 shifted toward upfield from δC 75.1 to δC 19.4.
A4: We have corrected the mistake. Thank you.
Q5: At line 105: the authors named the compound 2 as 6β,7β,14α,15β-tetrahydroxyl. However, in structure on Figure 1 the hydroxyl group at C-14 was resrepresented with beta-configuration. Check carefully the given names for compounds.
A5: We have checked and corrected this contents. Thank you.
Q6: Why the glucose moiety is not bonded to C-14 in structure 2?
A6: The key HMBC correlations from H-1’ to C-1 unambiguously confirmed that the sugar residue was located at C-1, not C-14. The NOESY correlations of H-1’/H-1 and H-1’/H2a was further supported this deduction.
Q7: Despite the reference cited in acid hydrolysis, the methodology employed should be summarized and added in experimental section.
A7: We added specific experimental methods of acid hydrolysis in experimental section. Thank you for your advice.

Reviewer 2 Report
English language needs substantial improvement. In some sentences, it is hard to understand what the authors mean. Verbs are missing.
The authors describe the isolation of 4 diterpenes from Isodon henryi plant. The identification was based mainly on NMR data and comparison with the known compound 5. ECD calculations for compound 1 and experimental ECD for compounds 1-4 were also performed, but these results are not discussed enough in the text. Could the authors compare with that of 5?
The specific rotations and melting points of isolated compounds must be added. Also, they must provide X-ray analysis for one of the new compounds to confirm the absolute configuration of them.
Line 74: check the sentence “was located at C-15”? or C-11?
Line 94: upfield or downfield?
Line 70 (H2-3 proton) and line 81 (H-20a): use the same symbol to indicate the methylene protons (correlation from COSY and NOESY spectra).
Author Response
Dear Professor:
Thank you very much for your letter and giving us opportunity to revise the manuscript entitled “Four new ent-Kaurane diterpene glycosides from Isodon henryi” (molecules-519355). We have carefully studied the reviewer’s critical comments and thoughtful suggestions. Based on the comments we received, careful modifications have been made to the manuscript. All changes made to the text are in red so that they may be easily identified. If there are more questions, we are willing to revise it again.
Below you will find our point-by-point responses to the reviewers’ comments.
Thank you and all the reviewers again for the kind and helpful advice.
A list of changes and responses to reviewers are as follows:
Reviewer 2:
Q1: English language needs substantial improvement. In some sentences, it is hard to understand what the authors mean. Verbs are missing.
A1: We have polished the manuscript in language by a professional English editor's
(1): Some necessary spaces were inserted between the words, and some redundant spaces were deleted.
(2): The misused Verbs have been revised. For example, “Diterpenoids from Isodon plants potent cytotoxic activities” should be “Diterpenoids from Isodon plants possess potent cytotoxic activities” in line 31; and “analyze” was revised in line 40. Similar Verbs missing were revised also in line 46 and line 78.
Q2: The authors describe the isolation of 4 diterpenes from Isodon henryi plant. The identification was based mainly on NMR data and comparison with the known compound 5 ECD calculations for compound 1 and experimental ECD for compounds 1-4 were also performed, but these results are not discussed enough in the text. Could the authors compare with that of 5?
A2: We have polished the manuscript in language by a professional English editor's
(1): The four compounds are all powders. We have tried many times to obtain single crystals of the four compounds. But they all failed.
(2): After consulting the literature, we found that many compounds including diterpenoids (rosthominsin A, sarcomililatins A, sarcomililatins B, sarcomililatol) have their absolute configurations determined by ECD calculation method [1-6]. And for detailed ECD calculation results of compound 1, see supplementary material S37.
[1]. Zhan R. Studies on Chemical Constituents and Biological Activities of Three Species of Isodon[D]. University of chinese academy of sciences, 2013.
[2]. Zhao J C , Wang Y L , Zhang T Y , Chen, Z. J. , Yang, T. M. , Wu, Y. Y. , Sun, C. P. , Ma, X. C. , Zhang, Y. X. . Indole diterpenoids from the endophytic fungus Drechmeria sp. as natural antimicrobial agents[J]. Phytochemistry, 2018, 148:21.
[3]. Lei F , Xiu-Qing S , Tian-Tian H , Kong-Kai, Z. , Jin-Hai, Y. , Jin-Tong, S. , Jie Z. , Hua Z. . Two new polyketides from the roots of, Stemona tuberosa[J]. Fitoterapia, 2018, 129:150-153.
[4]. Yin X , Zhao F , Feng X , Li, J. , Yang, X. , Zhang, H. , Tu, P. , Chai, X. Four new spirobenzylisoquinoline, N- oxide alkaloids from the whole plant of Corydalis hendersonii[J]. Fitoterapia, 2018, 128:31-35.
[5]. Li, S., Ye, F., Zhu, Z., Huang, H., Mao, S., Guo, Y. Cembrane-type diterpenoids from the South China Sea soft coral Sarcophyton mililatensis[J]. Acta Pharm Sin B, 2018.
[6]. Grzegorz, Z. , Ewa, M. , Agnieszka, K. , Jiri, K. , Petr, B. , Malgorzata, B. . Structure of supramolecular astaxanthin aggregates revealed by molecular dynamics and electronic circular dichroism spectroscopy[J]. Physical Chemistry Chemical Physics, 2018:10.1039.
Q3: The specific rotations and melting points of isolated compounds must be added. Also, they must provide X-ray analysis for one of the new compounds to confirm the absolute configuration of them.
A3: Thank you for your advice. We have added the optical rotation of isolated compounds to the manuscript. The four compounds are all powders. We have tried many times to obtain single crystals of the four compounds. But they all failed. Therefore, the absolute configuration cannot be determined by X-ray diffraction.
Q4: Line 74: check the sentence “was located at C-15”? or C-11?
Q4: Thank you for your advice. We have revised it in the revision.
Q5: Line 94: upfield or downfield?
A5: Thank you for your advice. We have revised this mistake.
Q6: Line 70 (H2-3 proton) and line 81 (H-20a): use the same symbol to indicate the methylene protons (correlation from COSY and NOESY spectra).
A6: Thank you for your advice. We have revised this mistake.

Round 2
Reviewer 1 Report
All suggestions were implemented in the second version of the paper. Doubts about the correct position of the glucosyl moiety were clarified with the expansion of the spectra.
I recomend the publication of the paper.
Author Response
Dear Professor:
Thank you very much for your letter and giving us opportunity to revise the manuscript entitled “Four New ent-Kaurane Diterpene Glycosides from Isodon henryi” (molecules-519355). Thank you very much for your valuable comments, making our manuscript more perfect.
Thank you and all the reviewers again for the kind and helpful advice.
Reviewer 1:
Q1: All suggestions were implemented in the second version of the paper. Doubts about the correct position of the glucosyl moiety were clarified with the expansion of the spectra.
A1: Thank you very much for your valuable suggestions to make the manuscript more perfect.
Q2: I recomend the publication of the paper.
A2: Thank you for your efforts in the publication of the manuscript; Thank you for your approval of the manuscript.

Reviewer 2 Report
English language must be improved particularly in abstract, introduction and conclusions sections.
The authors wrote (lines 55, 91, 110, 122) that compounds 1,2,3,4 were isolated as “an amorphous powder”. However, in lines 201, 213,217,223, they wrote that these compounds are “white crystalline powder”. They have to correct these inconsistencies. If the products are crystalline powders, may provide their X ray analysis. In Tetrahedron Letters 2017 58, 3574-3578 (ref. 16) for crystallization of ent-kaurane diterpenes, an EtOH-Water solution was used. See also Phytochemistry 2019, doi org 10.1016/j. phytochem. 2018.10.017.
Melting points must be added.
Optical rotations. The concentrations used are too low to trust the result. MeOD, a deuterated solvent, is not a common solvent for optical rotations measurements. Is there any sense giving three decimal points?
Author Response
Dear Professor:
Thank you very much for your letter and giving us opportunity to revise the manuscript entitled “Four new ent-Kaurane diterpene glycosides from Isodon henryi” (molecules-519355). We have carefully studied the reviewer’s critical comments and thoughtful suggestions. Based on the comments we received, careful modifications have been made to the manuscript. All changes made to the text are in blue so that they may be easily identified. If there are more questions, we are willing to revise it again.
Below you will find our point-by-point responses to the reviewers’ comments.
Thank you and all the reviewers again for the kind and helpful advice.
A list of changes and responses to reviewers are as follows:
Reviewer 2:
Q1: English language must be improved particularly in abstract, introduction and conclusions sections.
A1: Thank you very much for your comments. We have used professional English editing services to edit the full text (MDPI Author Services: english-edited-10550).
Q2: The authors wrote (lines 55, 91, 110, 122) that compounds 1,2,3,4 were isolated as “an amorphous powder”. However, in lines 201, 213,217,223, they wrote that these compounds are “white crystalline powder”. They have to correct these inconsistencies. If the products are crystalline powders, may provide their X ray analysis. In Tetrahedron Letters 2017 58, 3574-3578 (ref. 16) for crystallization of ent-kaurane diterpenes, an EtOH-Water solution was used. See also Phytochemistry 2019, doi org 10.1016/j. phytochem. 2018.10.017.
A2: Thank you very much for your valuable comments. I am very sorry for this mistake,and four compounds are amorphous powder. And we have modified the error in lines 201, 213, 217, 223.
As you mentioned, we also obtained crystalline powder of diterpene aglycon in the early stage (Dai, L.P.; Li, C.et.al. Molecules. 2015, 20, 17544-17556). And successfully obtained its single crystal (Dai, L.P, Zhang, L.X et.al. Nat. Prod. Res. 2018, 32, 2424-2430.)
However, this manuscript reports four compounds are diterpene glycosides. And all four compounds are amorphous powders. We have tried many times to obtain single crystals of four compounds. Regrettably, our efforts have failed.
Q3: Melting points must be added.
A3: Thank you for your advice. 4 compounds are amorphous powder, not crystalline powder.
Q4: Optical rotations. The concentrations used are too low to trust the result. MeOD, a deuterated solvent, is not a common solvent for optical rotations measurements. Is there any sense giving three decimal points?
A4: Thank you very much for your suggestion, we have re-measured optical rotations of 4 compounds according to your suggestion and revised them in the revision.
